# Tumor Necrosis Factor-Related Apoptosis-Inducing Ligand (TRAIL): A Novel Biomarker for Prognostic Assessment and Risk Stratification of Acute Pulmonary Embolism

**DOI:** 10.3390/jcm11133908

**Published:** 2022-07-05

**Authors:** Haixu Yu, Wei Rong, Jie Yang, Jie Lu, Ke Ma, Zhuohui Liu, Hui Yuan, Lei Xu, Yulin Li, Zhi-Cheng Jing, Jie Du

**Affiliations:** 1Beijing Anzhen Hospital of Capital Medical University and Beijing Institute of Heart Lung and Blood Vessel Diseases, Beijing 100029, China; yuhaixu6619@163.com (H.Y.); rongweiww@126.com (W.R.); yjyjlyang@163.com (J.Y.); jielu_0901@163.com (J.L.); make11191017@163.com (K.M.); lzh1540065015@163.com (Z.L.); 18911662931@189.cn (H.Y.); leixu2001@hotmail.com (L.X.); lyllyl_1111@163.com (Y.L.); 2Department of Cardiology and Institute of Vascular Medicine, Peking University Third Hospital, Beijing 100191, China; 3Department of Cardiology, Peking Union Medical College Hospital, Chinese Academy of Medical Sciences and Peking Union Medical College, Beijing 100730, China

**Keywords:** pulmonary embolism, TNF-related apoptosis-inducing ligand, prognosis, risk stratification

## Abstract

Background: Tumor necrosis factor (TNF)-related apoptosis-inducing ligand (TRAIL) is associated with poor prognosis in cardiovascular diseases. However, the predictive value of TRAIL for the short-term outcome and risk stratification of acute pulmonary embolism (PE) remains unknown. Methods: This study prospectively included 151 normotensive patients with acute PE. The study outcome was a composite of 30-day adverse events, defined as PE-related death, shock, mechanical ventilation, cardiopulmonary resuscitation, and major bleeding. Results: Overall, nine of 151 (6.0%) patients experienced 30-day adverse composite events. Multivariable logistic regression showed that TRAIL was an independent predictor of study outcome (OR 0.19 per SD; 95% CI 0.04–0.90). An ROC curve revealed that TRAIL’s area under the curve (AUC) was 0.83 (95% CI 0.76–0.88). The optimal cut-off value for TRAIL was 18 pg/mL, with a sensitivity, specificity, negative predictive value, positive predictive value, positive likelihood ratio, and negative likelihood ratio of 89%, 69%, 99%, 15%, 2.87, and 0.16, respectively. Compared with the risk stratification algorithm outlined in the 2019 ESC guidelines, our biomarker-based risk stratification strategy (combining TRAIL and hs-cTnI) has a similar risk classification effect. Conclusion: Reduced plasma TRAIL levels predict short-term adverse events in normotensive patients with acute PE. The combination of the 2019 ESC algorithm and TRAIL aids risk stratification in normotensive patients with acute PE.

## 1. Introduction

Venous thromboembolism (VTE), including deep vein thrombosis and pulmonary embolism, contributes a significant burden on health and survival and ranks third among life-threatening cardiovascular diseases [1]. Acute pulmonary embolism (PE) is the most severe clinical manifestation of VTE. Most patients with acute PE are normotensive, and early mortality ranges from 3–7% [2,3,4]. Early prognostic assessment and risk stratification for normotensive patients with acute PE is essential for determining appropriate treatment management approaches. The 2019 European Society of Cardiology (ESC) guidelines suggested that the extensively validated and broadly used simplified pulmonary embolism severity index (sPESI), combined with right ventricular (RV) dysfunction and laboratory biomarkers, can be used to classify acute PE patients without hemodynamic instability into intermediate- or low-risk groups. In addition to clinical parameters and scores, patients in the intermediate-risk group who display RV dysfunction and elevated cardiac troponin levels are classified into the intermediate-high-risk category [5]. Previous evidence demonstrated that a subgroup of normotensive patients with acute PE (i.e., intermediate-risk group) might benefit from aggressive treatment strategies [6]. Thus, optimizing risk stratification in normotensive PE is essential to enhance clinical practice.

Tumor necrosis factor (TNF)-related apoptosis-inducing ligand (TRAIL), which is also known as Apo-2 ligand (Apo-2L) or TNF superfamily 10 (TNFSN10), is a member of the TNF superfamily of cytokines, which is broadly expressed in various tissues of the human body [7]. TRAIL is selectively expressed in vascular smooth muscle cells of the pulmonary artery and aorta [8]. Soluble TRAIL mainly appears to be released by activated leukocytes such as monocytes and neutrophils [9]. TRAIL is a pro-apoptotic protein which has broad biological functions. TRAIL may play a crucial role in the pathway linking coagulation and inflammation elicited by thrombin and mediates the amplification of pro-coagulant endothelial microparticles released by thrombin and the inflammatory process [10]. Several clinical studies have shown that reduced TRAIL levels are associated with poor prognosis in patients with acute myocardial infarction or heart failure, suggesting that TRAIL has predictive effects in cardiovascular diseases [11,12,13].

In this study, we hypothesized that TRAIL may be involved in the pathophysiological mechanism of PE through the interplay between coagulation and inflammation and might assist in the prognostic assessment of patients with acute PE. Thus, our study aimed to identify the short-term prognostic assessment and risk stratification of TRAIL in normotensive patients with acute PE.

## 2. Materials and Methods

### 2.1. Study Design and Setting

We conducted a prospective study of normotensive patients with acute pulmonary embolism from 2015 to 2017 at Beijing Anzhen Hospital in China (NCT 04118634). Based on the amended Declaration of Helsinki, the study protocol was approved by the Ethics Committee of Beijing Anzhen Hospital (No. 2018048X), and all patients provided written informed consent.

### 2.2. Selection of Participants

As shown in Figure 1, normotensive patients (defined as SBP ≥ 90 mmHg) were consecutively enrolled if they had acute PE, were aged ≥ 18 years, and the onset of the illness was ≤14 days ago. Patients with acute PE were objectively confirmed by computed tomography pulmonary angiography (CTPA) and a ventilation-perfusion lung scan. The exclusion criteria were the following: [14,15,16] (1) hemodynamic instability: (A) cardiac arrest: cardiopulmonary resuscitation required; (B) obstructive shock: systolic blood pressure (BP) < 90 mmHg or vasopressors required to achieve a BP ≥ 90 mmHg despite adequate filling status and end-organ hypoperfusion (altered mental status; cold, clammy skin; oliguria/anuria); (C) persistent hypotension: systolic BP < 90 mmHg or systolic BP drop ≥ 40 mmHg lasting longer than 15 min and not caused by new-onset arrhythmia, hypovolaemia, or sepsis; (2) recurrence of PE; (3) chronic thromboembolic pulmonary hypertension; (4) life expectancy <3 months (i.e., the end stage of diseases); (5) ongoing pregnancy; (6) renal insufficiency (estimated glomerular filtration rate <30 mL/ min*1.73 m^2^) or hepatic dysfunction (Child–Pugh class B or C); (7) withdrawal of written consent for participation in this study; and (8) missing blood samples and troponin data.

### 2.3. Methods of Measurement

The diagnosis of acute PE was assessed using the Wells clinical probability rule, D-dimer, and imaging tests by the diagnostic algorithm outlined in the 2019 ESC guidelines [5]. All patients underwent transthoracic echocardiography within 24 h after diagnosis of PE. The diagnosis of RV dysfunction was based on the following diagnostic criteria [5]: (1) RV dilatation at the apical four-chamber view (RV end-diastolic diameter/left ventricular end-diastolic diameter >1.0), (2) depressed contractility of the RV free wall, (3) tricuspid regurgitation velocity acceleration, and (4) decreased tricuspid annular systolic excursion (<17 mm). The electronic medical record system obtained other clinical data, laboratory findings, and treatment details. According to the risk stratification strategy proposed in the 2019 ESC guidelines, all normotensive patients with acute PE were classified into the intermediate-high-, intermediate-low-, and low-risk groups according to their sPESI score, RV dysfunction, and troponin level. The physicians made treatment decisions while being unaware of TRAIL levels after carefully considering each patient’s clinical symptoms, laboratory findings, and imaging tests.

Venous plasma samples were collected from patients within 24 h after admission in vacuum tubes and immediately frozen at −80 °C after centrifugation at 3000× *g* for 10 min. Plasma TRAIL concentrations were determined using an ELISA kit (Ray Biotech, Inc. Norcross, GA, USA). Other laboratory tests were completed by the laboratory department of Beijing Anzhen Hospital. 

### 2.4. Outcome Measures

The study outcome was 30-day adverse composite events, defined as PE-related death or at least one of the following complications: (1) the need for mechanical ventilation assistance, (2) the need for catecholamine administration for treatment or prevention, (3) cardiopulmonary resuscitation, or (4) major bleeding. PE-related death was determined by (1) autopsy, (2) clinically severe acute PE, and (3) in cases where other causes were excluded. Major bleeding was defined as clinically overt bleeding accompanied by at least one of the following: (1) fatal bleeding or bleeding that occurred at critical sites or organs (intracranial, intraspinal, retroperitoneal, intraocular, and pericardial bleeding); (2) hemodynamic instability due to bleeding and/or a fall in the hemoglobin level ≥20 g/L, or bleeding that led to the transfusion of at least two units of blood [17]. 

All patients were followed up by pre-trained research staff. We determined the occurrence of the study outcome by using data collected through a review of the electronic medical records, clinical visits, and telephone follow-up interviews for up to 30 days.

### 2.5. Biomarker-Based Risk Algorithm

In the 2019 ESC prognostic strategy, risk assessment for early mortality consists of seven clinical parameters (sPESI rule), two relevant imaging modalities (TTE or CTPA), and four cardiac biomarkers (troponin, NT-proBNP, H-FABP, and copeptin). Objective assessments are relatively time-consuming, labor-intensive, and cost-intensive. Thus, in this study, a biomarker-based risk algorithm was developed to evaluate the risk assessment of normotensive patients with acute PE. This biomarker-based stratification strategy was established using TRAIL combined with hs-cTnI levels. According to previous studies [18,19,20], hs-cTnI possessed superior negative predictive values (NPV) for short-term adverse events and could be used as the first step in risk stratification to classify patients with low-risk acute PE.

### 2.6. Statistical Analyses

The Kolmogorov–Smirnov test for normal distribution was used for continuous variables. Skewed continuous variables were expressed as medians (interquartile range [IQR]). Categorical variables were expressed as absolute numbers or percentages. Comparisons of continuous variables were analyzed using unpaired Student’s *t*-tests or Mann–Whitney *U* tests, and comparisons of categorical variables were analyzed using Chi-squared or Fisher’s exact tests. Correlations between continuous variables were analyzed using Spearman’s rank correlation coefficient. The prognostic relevance of clinical variables, cardiac biomarkers, TRAIL levels, and sPESI scores for 30-day adverse events was calculated using univariate (unadjusted) and multivariate (adjusted) logistic regression analysis, producing odds ratios (OR) and 95% confidence intervals (CIs). Factors for inclusion in the multivariate analysis were determined after considering the findings from previous publications and the latest ESC guidelines and significant predictors (*p* < 0.05) from the univariate analysis. Receiver operating characteristic (ROC) curve analysis was performed to determine the area under the curve (AUC) of TRAIL cut-off values for the study outcomes. Youden’s index was used to identify optimal cut-off values. Sensitivity, specificity, negative predictive values (NPV), positive predictive values (PPV), negative likelihood ratios (−LR), positive likelihood ratios (+LR), and the corresponding 95% CIs were calculated. The McNemar–Bowker test was used to compare the distribution of patients in different risk stratification strategies (2019 ESC algorithm and biomarker-based approach). Two-tailed *p* values < 0.05 were considered statistically significant. All statistical analyses were conducted using SPSS (version 25.0; IBM, Chicago, IL, USA). 

## 3. Results

### 3.1. Characteristics of Study Subjects

Between January 2015 and December 2017, 221 patients were screened, of whom 70 met the exclusion criteria (flow chart shown as Figure 1). Among the 151 patients who participated in this study, nine (6%) experienced 30-day adverse composite events. One patient died directly due to PE; seven patients required catecholamine administration for treatment or prevention. Two patients required mechanical ventilation, two required cardiopulmonary resuscitation, and one suffered major bleeding. The clinical and demographic characteristics of study participants with and without study events are presented in Table 1. The event group more frequently experienced syncope, RV dysfunction, higher BNP and hs-cTnI concentrations, and sPESI scores ≥ 1 compared to the non-event group. Additionally, nine (6.0%) patients received thrombolytic therapy and five (55.6%) experienced adverse outcomes.

### 3.2. Association between TRAIL Levels and Short-Term Prognosis

The median TRAIL concentration was 23.1 pg/mL (IQR 15.0–32.3) in all patients. Patients in the events group had significantly lower TRAIL levels (median, 10.1 pg/mL [IQR 3.6–16.4]) than patients in the non-event group (median 23.5 pg/mL [IQR 16.1–32.6], *p* = 0.001). The TRAIL concentrations were weakly correlated with BNP (r = −0.28, *p* = 0.001) and hs-cTnI (r = −0.24, *p* = 0.003). The predictors of 30-day adverse composite events were investigated using a univariate logistic regression analysis (Table 2). Significant predictors of 30-day adverse composite events in the univariate analysis included syncope (OR = 9.83; 95% CI 2.30–42.08, *p* = 0.002), RV dysfunction (OR = 16.5; 95% CI 3.82–71.30, *p* = 0.000), BNP (OR = 3.60 per SD; 95% CI 1.91–6.78, *p* = 0.000), TRAIL (OR = 0.18 per SD; 95% CI 0.06–0.56, *p* = 0.003), and a sPESI score ≥ 1 (OR = 16.17; 95% CI 1.95–133.11, *p* = 0.010). Considering the findings from previous publications and the latest ESC guidelines, significant predictors from the univariate analysis and cardiac troponin (hs-cTnI) were included in the multivariate logistic regression analysis (Table 2). After adjustment, TRAIL was independently and significantly associated with 30-day adverse composite events in normotensive patients with acute PE (OR = 0.19 per SD; 95% CI 0.04–0.90, *p* = 0.036). As shown in Figure 2, ROC analysis revealed that the AUC of TRAIL was 0.83 (95% CI 0.76–0.88, *p* < 0.001) for the prediction of short-term adverse outcomes, and the optimal cut-off value for TRAIL based on Youden’s index was 18 pg/mL, at which point the sensitivity, specificity, NPV, PPV, +LR, and −LR were 89%, 69%, 99%, 15%, 2.87, and 0.16, respectively. 

### 3.3. TRAIL’s Role in Risk Stratification 

According to the 2019 ESC risk algorithm (Figure 3), 10 (6.6%) patients were classified into the intermediate-high risk group, 78 (51.7%) into the intermediate-low risk group, and 63 (41.7%) into the low-risk group. During the follow-up, the 30-day adverse composite events occurred in 5 (50%), 4 (5.1%), and 0 (0%) patients, respectively. The risk assessment using the biomarker-based strategy based on hs-cTnI and TRAIL is shown in Figure 3. As with the 2019 ESC risk algorithm, the stepwise biomarker-based strategy demonstrated strong predictive performance in identifying intermediate-high- and low-risk group patients (Table 3). Both the biomarker-based strategy and the 2019 ESC algorithm showed high sensitivity (100%) and NPV (100%) in identifying low-risk patients, while the biomarker-based strategy had higher specificity than the 2019 ESC algorithm (65% vs. 44%, *p* < 0.001). When identifying intermediate-high-risk group patients, both strategies had high specificity (88% vs. 96%, *p* < 0.001) and the biomarker-based strategy had a superior trend of sensitivity (89% vs. 56%, *p* = 0.375). To combine the performance of the biomarker-based strategy and the 2019 ESC algorithm, we tested whether TRAIL may improve patients re-classified as belonging to the intermediate-high risk group, as shown in Figure 4. Using TRAIL < 18 pg/mL to further stratify patients in the intermediate-low risk group, 28 patients were identified as being at higher risk, with four adverse events. The prognostic performance of risk assessment using the 2019 ESC algorithm and TRAIL for the prediction of an adverse 30-day outcome is shown in Table 3, for which the sensitivity, specificity, NPV, PPV, +LR, and −LR were 100%, 80%, 100%, 24%, 5, and 0, respectively.

## 4. Discussion

This study investigated the relationship between plasma TRAIL concentrations and short-term adverse outcomes and whether TRAIL can optimize the current risk stratification. Using a cut-off value of 18 pg/mL, we found that decreased plasma TRAIL levels had an independently prognostic performance for 30-day adverse outcomes. A stepwise biomarker-based risk assessment strategy combining hs-cTnI and TRAIL improves predictive performance in identifying intermediate-high- and low-risk group patients. The combination of the 2019 ESC algorithm and TRAIL aids risk stratification in normotensive patients with acute PE.

### 4.1. The Potential Role of TRAIL in PE

TRAIL exists as either a type II membrane protein or a soluble protein. TRAIL receptors are expressed in the cardiovascular system in vascular smooth cells and cardiomyocytes, including osteoprotegerin (OPG). TRAIL has been found to play a role in ischemic vascular diseases and cardiovascular disease (CVD) [20,21,22,23,24]. Several prospective studies have demonstrated that lower TRAIL concentrations predicted poor prognosis in patients with CVD [13,25,26]. In our study, lower TRAIL concentrations were associated with short-term adverse outcomes. Low levels of TRAIL tend to represent poor prognosis. This is similar to the findings of several previous studies, in which serum TRAIL levels were negatively related to the severity of coronary heart disease [27], lower serum TRAIL levels were associated with worse outcomes in patients with acute myocardial infarction [28], and higher TRAIL levels in patients with advanced heart failure were associated with an improved prognosis [12,29]. Despite this, it is unclear how TRAIL can clinically influence the thrombosis and inflammation process during acute PE. However, it is plausible that the interaction between TRAIL and its receptors modulates the progression of thromboembolism. The role of inflammation-modulating maladaptive RV remodeling and dysfunction has been demonstrated. Acute PE leads to a cascade of inflammatory response which might be followed by leukocyte recruitment to the lesion. TRAIL recruits activated leukocytes to a particular tissue and initiates apoptosis to terminate the immune response. TRAIL promotes the proliferation of vascular smooth muscle cells and neovascularization [28,29]. TRAIL also enhances endothelial nitric oxide synthase phosphorylation, NOS activity, and NO synthesis; thus, it causes vasodilation [30,31]. Interestingly, there is a negative correlation between TRAIL and hsCRP, which provides further support for the protective role of TRAIL in the development of atherosclerosis and acute coronary disease [32].

### 4.2. The Combination of TRAIL and the 2019 ESC Algorithm for Risk Assessment in Normotensive Patients with Acute PE

Based on the 2019 ESC guidelines, treatment decisions for normotensive patients with acute PE need to be based on a risk stratification strategy, with low-risk patients being considered for early discharge and home treatment, intermediate-low or intermediate-high risk patients being closely monitored and offered reperfusion therapy if deterioration occurs. Recent cohort studies developed combination models for the identification of intermediate-high-risk PE patients (e.g., PREP score, FAST score, and Bova score) [33,34,35], and several studies investigated the prognostic value of biomarkers on risk stratification (e.g., Copeptin and Lipocalin-2) [16,19]. Due to the relatively limited performance of the 2019 ESC algorithm, we developed a novel and simple stepwise biomarker-based strategy using TRAIL and hs-cTnI. More patients were re-classified into the low-risk and intermediate-high risk groups using a biomarker-based algorithm. To combine the performance of the biomarker-based strategy and the 2019 ESC algorithm, we also tested whether TRAIL may improve patients re-classified as belonging to the intermediate-high risk group. As shown in Table 3 and Figure 4, the prognostic performance of risk assessment was improved using the 2019 ESC algorithm and TRAIL to predict an adverse 30-day outcome.

There are some limitations in this study that merit mentioning here. First, the included population came from a single center, and the number of people who experienced an outcome was low. However, adverse outcomes (6%) were similar to those reported in other studies [14,15,18,19]. Second, of the 221 patients screened, 31 (14.0%) were excluded due to missing data. Given the small size and the low event rate of this study, we could not evaluate if TRAIL has additional value on top of the existing risk stratification. Further large-scale studies are required in future using independent study cohorts. This study also lacked multiple consecutive measurements for TRAIL. The mechanism and pathophysiological process throughout the pulmonary embolism need to be further explored and validated.

## 5. Conclusions

In conclusion, reduced plasma TRAIL levels predict short-term adverse events in normotensive patients with acute PE. The combination of the 2019 ESC algorithm and TRAIL aids risk stratification to assist physicians in the making of treatment decisions and care of patients.

## Figures and Tables

**Figure 1 jcm-11-03908-f001:**
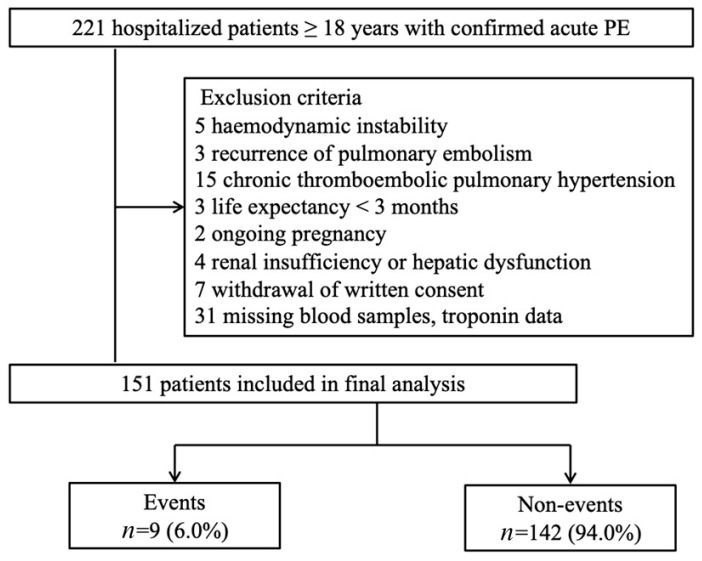
Study participants flow diagram. PE, pulmonary embolism.

**Figure 2 jcm-11-03908-f002:**
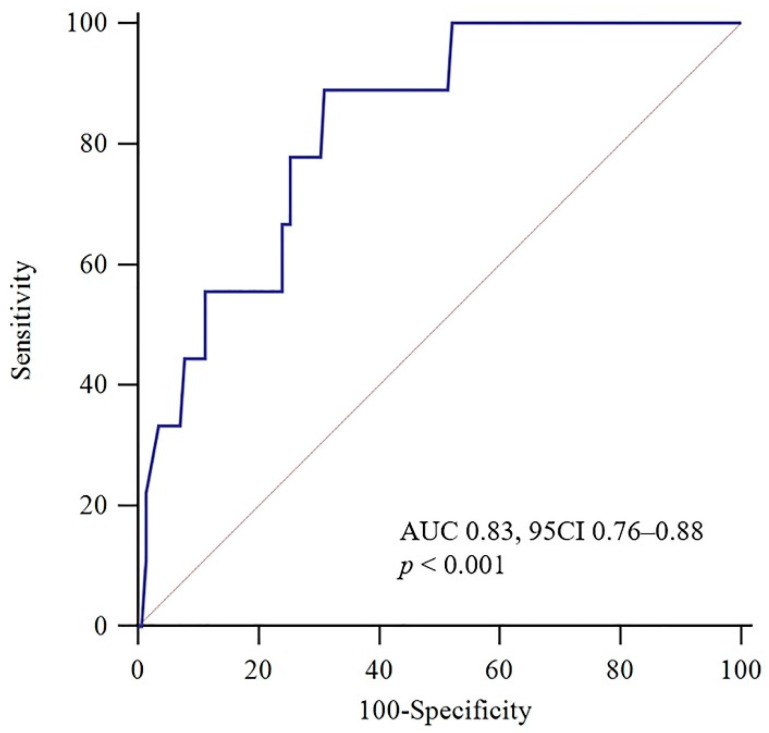
Receiver operating characteristic (ROC) curve for TRAIL concerning an adverse 30-day outcome. AUC: area under the curve; CI: confidence interval.

**Figure 3 jcm-11-03908-f003:**
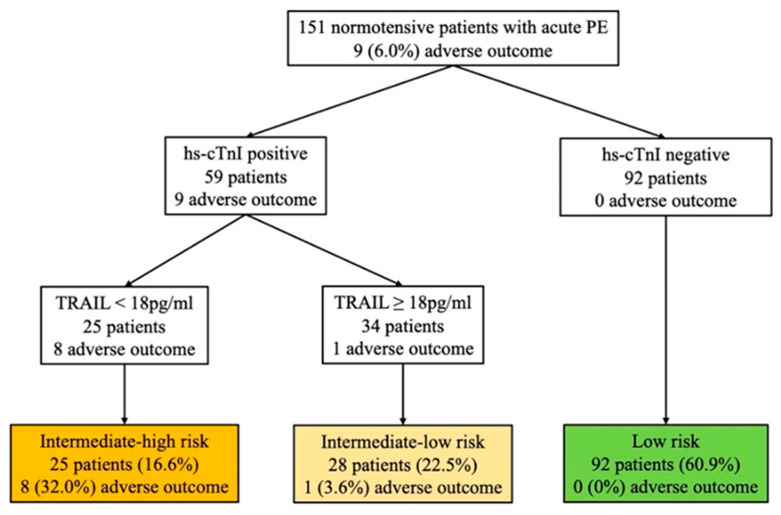
Risk assessment using the biomarker-based strategy based on hs-cTnI and TRAIL. The number (%) of patients with an adverse 30-day outcome is shown for each strategy. Hs-cTnI levels >0.04 ng/mL are defined as positive. PE: pulmonary embolism; hs-cTnI, high-sensitivity cardiac troponin I; TRAIL, tumor necrosis factor-related apoptosis-inducing ligand.

**Figure 4 jcm-11-03908-f004:**
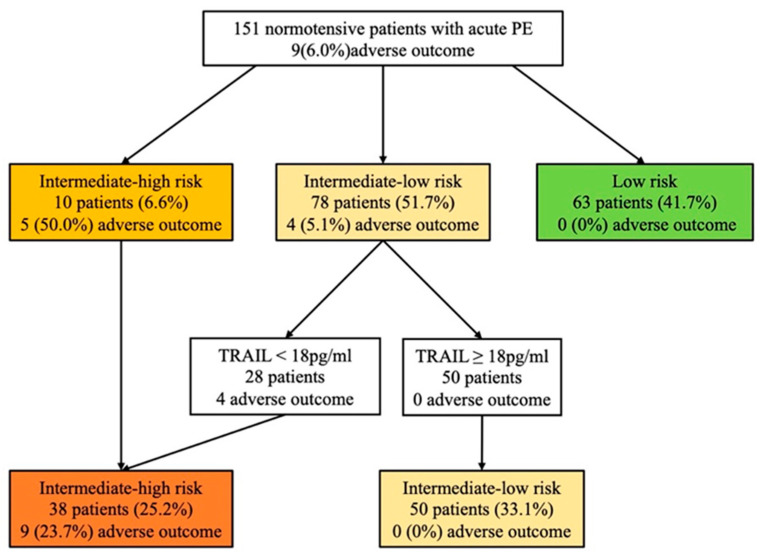
Risk assessment using the 2019 ESC algorithm and TRAIL. The number (%) of patients with an adverse 30-day outcome is shown for each strategy. Hs-cTnI levels > 0.04 ng/mL are defined as positive. PE: pulmonary embolism; TRAIL, tumor necrosis factor-related apoptosis-inducing ligand.

**Table 1 jcm-11-03908-t001:** Baseline characteristics of normotensive patients with acute pulmonary embolism.

	All Patients (*n* = 151)	Non-Events (*n* = 142)	Events (*n* = 9)	*p* Value
**Age, years**	66 (60–73)	66 (60–73)	62 (48–72)	0.453
**Male**	63 (41.7)	60 (42.3)	3 (33.3)	0.735
**Risk factors for VTE**				
History of VTE	19 (12.6)	19 (13.4)	0	0.603
Immobility	13 (8.6)	12 (8.5)	1 (11.1)	0.566
Recent surgery	8 (5.3)	7 (4.9)	1 (11.1)	0.396
Recent long travel	2 (1.3)	2 (1.4)	0	1.000
Recent fracture	9 (6.0)	8 (5.6)	1 (11.1)	0.434
**Comorbidities**				
Cancer	9 (6.0)	9 (6.3)	0	1.000
COPD	8 (5.3)	7 (4.9)	1 (11.1)	0.396
Coronary heart disease	25 (16.6)	1 (11.1)	24 (16.9)	1.000
**Symptoms and signs**				
Chest pain	39 (25.8)	38 (26.8)	1 (11.1)	0.448
Dyspnea	139 (92.1)	130 (91.5)	9 (100.0)	1.000
Syncope	30 (19.9)	24 (16.9)	6 (66.7)	**0.002**
SBP, mmHg	124 (114.5–124)	124 (115–138)	120 (113–134)	0.691
SBP < 100 mmHg	4 (2.6)	3 (2.1)	1 (11.1)	0.220
Heart rate, bpm	82 (73–98)	82 (72–96)	97 (84–102)	**0.010**
Heart rate ≥ 110 bpm	9 (6.0)	7 (4.9)	2 (22.2)	0.092
SaO_2_ < 90%	15 (9.9)	13 (9.2)	2 (22.2)	0.220
Elevated PASP	49 (32.5)	45 (31.7)	4 (44.4)	0.473
RV dysfunction (on TTE)	15 (9.9)	10 (7.0)	5 (55.6)	**0.001**
LVEF, %	63 (60–67)	64 (60–68)	60 (56–64)	0.083
**Laboratory biomarkers**				
D-Dimer, ng/mL	2166 (1076–3134)	2114 (1056–3110)	2823 (2389–3134)	0.088
Creatinine, µmol/L	73.5 (61.1–83.7)	73.2 (60.5–83.8)	75.0 (62.6–83.1)	0.75
BNP, pg/mL	141 (46–364)	118 (44.0–310.0)	1000 (653–2054)	**0.001**
hs-cTnI, ng/mL	0.03 (0.01–0.15)	0.02 (0.01–0.11)	0.27 (0.09–0.91)	**0.001**
TRAIL, pg/mL	23.1 (15.0–32.3)	23.5 (16.1–32.6)	10.1 (3.6–16.4)	**0.001**
**sPESI ≥ 1**	55 (36.4)	47 (33.1)	8(88.9)	**0.001**
**Treatment**				
Thrombolytic therapy	9 (6.0)	4 (2.8)	5 (55.6)	**0.000**

Data are presented as median (interquartile range) or number (%). VTE, venous thromboembolism; COPD, chronic obstructive pulmonary disease; SBP, systolic blood pressure; bpm, beats per minute; SaO_2_, arterial oxyhemoglobin saturation; PASP, pulmonary artery systolic pressure; RV, right ventricular; TTE, transthoracic echocardiography; LVEF, left ventricular ejection fraction; BNP, brain natriuretic peptide; hs-cTnI, high-sensitivity cardiac troponin I; TRAIL, tumor necrosis factor-related apoptosis-inducing ligand; sPESI, simplified Pulmonary Embolism Severity Index.

**Table 2 jcm-11-03908-t002:** Predictors of an adverse 30-day outcome.

	OR	95%CI	*p* Value
Univariable analysis ^a^			
Age > 80 years	3.43	0.36–32.90	0.286
Cancer	-	-	-
COPD	2.41	0.26–22.05	0.436
Syncope	9.83	2.30–42.08	0.002
SBP < 100 mmHg	5.79	0.54–62.12	0.147
Heart rate ≥ 110 bpm	5.51	0.96–31.57	0.055
SaO_2_ < 90%	2.84	0.53–15.09	0.222
RV dysfunction (on TTE)	16.5	3.82–71.30	0.000
BNP, pg/mL, per SD	3.60	1.91–6.78	0.000
hs-cTnI, ng/mL, per SD	1.25	0.85–1.85	0.254
TRAIL, pg/mL, per SD	0.18	0.06–0.56	0.003
sPESI ≥ 1	16.17	1.96–133.11	0.010
Multivariable analysis			
Syncope	2.48	0.20–31.12	0.481
RV dysfunction (on TTE)	16.47	1.06–256.27	0.045
BNP, pg/mL, per SD	3.68	1.24–10.89	0.019
hs-cTnI, ng/mL, per SD	1.45	0.64–3.32	0.375
TRAIL, pg/mL, per SD	0.19	0.04–0.90	0.036
sPESI ≥ 1	1.09	0.06–21.54	0.956

OR, odds ratio; SD, standard deviation; COPD, chronic obstructive pulmonary disease; SBP, systolic blood pressure; bpm, beats per minute; SaO_2_, arterial oxyhemoglobin saturation; RV, right ventricular; TTE, transthoracic echocardiography; BNP, brain natriuretic peptide; hs-cTnI, high-sensitivity cardiac troponin I; TRAIL, tumor necrosis factor-related apoptosis-inducing ligand; sPESI, simplified Pulmonary Embolism Severity Index. ^a^ Variables found to significantly predict an adverse 30-day outcome in the univariate analysis are displayed. Additionally, hs-cTnI levels and all variables included in the sPESI are shown. The logistic regression analysis calculates odds ratios (ORs) and their respective 95% confidence intervals (CIs) for an adverse 30-day outcome.

**Table 3 jcm-11-03908-t003:** Prognostic performance of risk assessment strategies for the prediction of an adverse 30-day outcome.

	Biomarker-Based Algorithm (95% CI)	2019 ESC Algorithm (95% CI)	Combination of TRAIL and the 2019 ESC Algorithm (95% CI)
Low-risk vs. intermediate-low- and intermediate-high-risk			
Sensitivity, %	100 (66–100)	100 (66–100)	100 (66–100)
Specificity, %	65 (56–73)	44 (36–53)	80 (72–86)
PPV, %	15 (13–18)	10 (9–12)	24 (18–30)
NPV, %	100	100	100
+LR	2.84 (2.3–3.5)	1.80 (1.6–2.1)	5 (3.5–6.8)
−LR	0	0	0
Low-risk and intermediate-low- vs. intermediate-high-risk			
Sensitivity, %	89 (52–100)	56 (21–86)	-
Specificity, %	88 (82–93)	96 (92–99)	-
PPV, %	32 (22–44)	50 (26–74)	-
NPV, %	99 (95–99)	97 (94–99)	-
+LR	7.42 (4.5–12.3)	15.78 (5.6–44.7)	-
−LR	0.13 (0.02–0.8)	0.46 (0.2–1.0)	-

ESC, european society of cardiology; CI, confidence interval; TRAIL, tumor necrosis factor (TNF)-related apoptosis-inducing ligand; PPV, positive predictive values; NPV, negative predictive values; +LR, positive likelihood ratios; -LR, negative likelihood ratios.

## Data Availability

The data underlying the results presented in the study are available from Beijing Anzhen Hospital.

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
