# Peer review of "Tumor Necrosis Factor-Related Apoptosis-Inducing Ligand (TRAIL): A Novel Biomarker for Prognostic Assessment and Risk Stratification of Acute Pulmonary Embolism"

_jcm, 2022, doi:10.3390/jcm11133908_

Round 1
Reviewer 1 Report
Very interesting paper. You should be commended for your work. The small event size is a limitation as there is an inability to match the factors which are known to have bad outcomes - high HR, RV dysfunction, syncope as well as other biomarkers which are more readily available troponin or BNP. This lead the question that you maybe using a less available biomarker to identify the same patient population.
I think adding to the ESC is okay but this is not how people would use this biomarker in reality. You should use it to add to the idea of pathophysiology more than prognostication. Maybe even in short term response to therapy?
Author Response
Thanks to the reviewer for this suggestion. This study sample was limited by the small size of the population (limitation paragraph). And based on this observational study, our study group is preparing further pathophysiological investigation and an independent validation cohort. As for the clinical application, we also need further evaluation in a larger population and the associated economic costs.

Reviewer 2 Report
The authors attempt to assess the short-term prognostic value and risk stratification of TRAIL in normotensive patients with acute PE, but there are significant issues that should be addressed. This is a positive study driven by catecholamine administration, a rather weak endpoint, that should be elucidated.
Although CTEPH was an exclusion criterion, 33% of pts had pulmonary hypertension as a comorbidity. The authors should comment on this, as well as on the cause of this pulmonary hypertension. If this is left heart disease pulmonary hypertension (a condition in which TRAIL is severely affected), then this could be a methodological flaw of the study.
Line 82: 'Onset of illness ≤ 14 days' : Is illness the PE? If so, this sounds as long time interval after PE, especially in a study with the the main outcome is the 30-day adverse composite events. This is a limitation of the study in a cohort of patients with not severe PH. Did patients with more recent PE had any difference compared to those with older (after the first wk) PE?
Line 113: 'Venous plasma samples were collected from patients within 24 hours after admission'. How does this combine with onset of illness<14d?
Line 121: 'the need for catecholamine administration for treatment or prevention' : How is prevention defined? Did the treating physicians administer cathecholamines based on a certain algorithm or arterial pressure level, or on arbitrary basis? If the latter is the case, how could this serve as outcome? The positive results are driven by catecholamine administration (7cases), so details no dosis and duration to assess degree of hemodynamic compromise should be provided.
4pts had SBP<100mmHg, but were considered normotensive. How was normal blood pressure defined?
20% of normotensive pts had syncope. Any explanation for this high incidence? Any association of syncope with TRAIL levels? Was it PE related? Did these pts receive thrombolysis?
Do the authors have any data on the kinetics of TRAIL? Risk assessment based on single measurements could be a limitation. The authors should comment on the independent factors that could affect TRAIL levels.
The discussion should focus more on the findings, it is rather theoretical.
